# *Wolbachia* and its pWCP plasmid show differential dynamics during the development of *Culex* mosquitoes

Alice Brunner,[1] Camille Gauliard,[1] Jordan Tutagata,[1] Seth R. Bordenstein,[2] Sarah R. Bordenstein,[2] Blandine Trouche,[1] Julie Reveillaud[1]

**ABSTRACT** Mosquitoes are major vectors of pathogens such as arboviruses and parasites, causing significant health impacts each year. *Wolbachia*, an intracellular bacterium widely distributed among arthropods, represents a promising vector control solution. This bacterium can reduce the transmission of dengue, Zika, and chikungunya arboviruses and manipulate the reproduction of its host through its prophage WO. Although research on the *Wolbachia* mobilome primarily focuses on WO and the phenotypes it induces, the function of *Wolbachia* plasmid pWCP, recently discovered and reported to be strikingly conserved worldwide, remains unknown. In this study, we analyzed the presence and abundance of pWCP as well as *Wolbachia* in two different species of *Culex* mosquitoes, one of the most widespread genera in the world and a vector of numerous diseases. We compared the relative densities of the bacterium and its mobile genetic element in *Culex pipiens molestus* and *Culex quinquefasciatus*, a facultatively autogenous and an anautogenous species, respectively, throughout their development from the larval stage L1 to the adult individual specimen using quantitative Polymerase Chain Reaction (PCR). Our results suggest that 2–5 copies of pWCP occur in *Wolbachia* cells on average, and the plasmid co-replicates with *Wolbachia* cells. Moreover, *Wolbachia* and pWCP exhibit differential levels of abundance at specific development stages throughout the mosquito's life cycle in each species. These findings indicate important, and likely beneficial, roles for the plasmid in the bacterium's biology in different mosquito species as well as complex interaction dynamics between *Wolbachia* and its host during its life cycle.

**IMPORTANCE** Mosquitoes of the *Culex* genus are critical vectors for numerous diseases, causing significant public health concerns. The intracellular bacterium *Wolbachia* has emerged as a promising vector control solution due to its ability to interfere with pathogen transmission and manipulate mosquito reproduction. However, unlike the extensively studied WO phage, the biological significance and function of *Wolbachia's* pWCP plasmid, a recently discovered and strikingly conserved mobile genetic element in *Culex* species, remain unknown. This study investigates the developmental dynamics of pWCP and *Wolbachia* in two *Culex* mosquito species, *Culex pipiens molestus* and *Culex quinquefasciatus* across their life cycle. In general, the abundance levels of *Wolbachia* and the plasmid were found to vary across life stages and differ between the two species. However, a relatively small number of pWCP copies were observed per *Wolbachia* cell, together with a co-replication of the plasmid with the bacterium for most developmental stages. Altogether, these findings suggest a likely beneficial and non-parasitic role for pWCP in *Wolbachia's* biology, which may contribute to the intricate interactions between the bacterium and its mosquito hosts.

**KEYWORDS** mosquitoes, *Wolbachia*, mobile genetic elements, plasmid, life cycle

**Peer Reviewer** Marco Salgado, University of Helsinki, Helsinki, Finland

Address correspondence to Julie Reveillaud, reveillaud.j@gmail.com.

The authors declare no conflict of interest.

See the funding table on p. 9.

*[This article was published on 31 March 2025 with an error in the title. The title was corrected in the current version, posted on 3 April 2025.]*

Mobile genetic elements (MGEs) are essential drivers of genomic plasticity and adaptation in organisms, notably by facilitating horizontal gene transfer (1–3). They include diverse entities such as insertion sequences (IS), transposons (Tn), integrons (In), phages, and plasmids, which have the ability to move or spread within the same genome or between different cells, thereby altering the genetic structure of their hosts (4–7). Through their various mechanisms of propagation, such as conjugation (mediated by plasmids and integrative conjugative elements), transduction (mediated by phages), and transformation (uptake of extracellular DNA), these MGEs actively contribute to the acquisition of new functions and the rapid evolution of microbial communities (8, 9).

Phages and plasmids, as major categories of MGEs, have a particularly significant impact on bacterial evolution (10, 11). In addition to their core genes, these MGEs typically carry accessory genes that provide selective advantages to their host cells, such as antibiotic resistance, virulence factors, or unusual metabolic pathways (12–14). Plasmids, which are present in all domains of life (15), generally exist as circular extrachromosomal DNA and replicate independently of the bacterial chromosomal DNA (16). To coexist stably with their hosts and minimize metabolic burden, plasmids must regulate their replication so that their copy number remains consistent within a given host and under defined growth conditions (17). However, recent studies show that modulating plasmid copy numbers can directly influence bacterial growth and have significant impacts on the metabolic costs borne by the host (18). For example, the variability in plasmid copy number can result in varied effects on the stability of replication, plasmid loss, or even impose a metabolic burden, thereby influencing bacterial interactions within the host cell (18).

In addition, recent advances in metagenomics, in particular state-of-the-art assembly and binning approaches, are shedding light on the diversity and dynamics of plasmids that lack identifiable traits or detectable host markers (19–21). For example, the cryptic plasmid pBI143 of *Bacteroides fragilis*, recently discovered through metagenomic studies of the human gut microbiota, was found as one of the most abundant elements in this ecosystem (22). Although its precise function remains unknown, the relative copy number of the element increases during stress, such as in inflammatory bowel disease (22). Similarly, the first identified plasmid of *Wolbachia*, named pWCP for *Wolbachia* plasmid in *Culex pipiens*, was found as highly conserved in *Culex* spp. and is present worldwide (23). Data showed a rather stable copy number of 4–5 in the ovaries of adult females worldwide (23), yet with some interindividual variations possibly reflecting distinct physiological states of the mosquito hosts (23). Overall, the discovery of pWCP could open new possibilities for effective genome-editing strategies in a bacterium that has, until now, been resistant to genetic modification.

*Wolbachia* was first identified in 1924 (24) and belongs to the order *Rickettsiales* (Alphaproteobacteria). The main evolutionary lineages of *Wolbachia* are referred to as "supergroups" and are designated by letters (A–F, H–K, and S) (25). Supergroups A and B are found in many terrestrial arthropods (26) and infect over 70% of these arthropods (27), including 50% of the insect species (28). Its prevalence and frequency within insect populations are explained by its ability to manipulate host reproduction (29) and possibly by nutritional mutualism (30).

*Wolbachia* can induce various reproductive manipulation phenotypes, notably cytoplasmic incompatibility (CI) (31, 32). CI causes sterility in crosses between infected males and uninfected females or between individuals infected with different and incompatible *Wolbachia* strains, facilitating the establishment of infected populations (33, 34). This phenomenon is used in biocontrol programs, such as the World Mosquito Program, based on the release of *Aedes aegypti* mosquitoes infected with *Wolbachia* strains to fight diseases like dengue and Zika (35–38). The effectiveness of this method is enhanced by *Wolbachia*'s ability to block pathogen transmission in mosquitoes (33, 39–41). Mosquitoes trans-infected with *Wolbachia* show varying levels of resistance to dengue, Zika, and chikungunya viruses, with viral inhibition often linked to *Wolbachia* density in somatic tissues (40).

CI is mediated by *cifA* and *cifB* genes located in the WO prophage of Wolbachia (42). Recent findings show that plasmids in *Wolbachia* and related bacteria, such as *Rickettsia*, carry *cif* gene homologs, indicating a potential role of horizontal plasmid transfer in CI acquisition (43, 44). Other plasmids, notably pWALBA1 and pWALBA2 in *Wolbachia* of *Aedes albopictus* (*w*AlbA), showed features rather similar to pWCP (43) with many unknown genes. The mobile genetic element pWCP comprises 9.23 kbp encoding 14 genes (45). Of these, seven have putative functions, including some involved in DNA replication (*DnaB*-like helicase), plasmid partitioning (*ParA*-like), toxin-antitoxin stability systems (*RelBE* loci), and a transposable element (IS110-family), which collectively suggest a role in plasmid maintenance and mobility. The remaining seven genes are of unknown function (45). It is possible that one specific or a set of pWCP genes plays a key role for the bacterium and the mosquito holobiont as a whole, particularly during physiological changes or insect metamorphosis, which might influence pWCP copy numbers in certain conditions. This is observed in the symbiont *Buchnera aphidicola* of aphids, where the copy number of the leucine biosynthesis plasmid increases under amino acid deprivation in the host (46). However, despite the importance of pWCP for *Wolbachia* in *Culex*, much remains unknown about its replication mode, behavior, and interaction with its host.

In this study, we explore the relative abundance of plasmid pWCP and *Wolbachia* itself throughout the development of two mosquito species of the *Culex* genus, *Cx. pipiens molestus,* and *Cx. quinquefasciatus,* a facultatively autogenous and anautogenous species, respectively. Mosquitoes were analyzed from the first larval stage to the fourth larval stage, pupae, and adults (male and female) stages using quantitative PCR to determine the variability or stability of the plasmid and *Wolbachia* across the different stages of development.

## MATERIALS AND METHODS

### Mosquito rearing and sampling

We maintained the *Cx. pipiens molestus* (Celestine strain, autogenous, originally from Montpellier, reared in the laboratory since October 2023, fifth generation) and *Cx. quinquefasciatus* (SLAB strain, initially isolated from the San Joaquin Valley, California, USA, in the 1960s, ca. 96th generation in our laboratory) at 27°C with 70% relative humidity and a 12–12 h light-dark cycle. We provided a 10% sugar water solution, prepared with white sugar dissolved in osmosis water, weekly in Erlenmeyer flasks for both strains.

The SLAB strain, being anautogenous, additionally received a monthly blood meal from canaries (*Serinus canaria*) restrained in the mosquito cages. We reared larvae from both species in insectary trays (30 × 20 × 6 cm) filled with 1 L osmosis water at a density of approximately 200 larvae per tray. The larval diet consisted of a mixture of one-third of Tetramin fish food (Tetra, Germany) and two-thirds of rabbit pellets (Versele-Laga, Belgium).

We collected 25 individuals at each developmental stage (L1, L2, L3, L4, pupa, adult female, and adult male) from each species (*n* = 175) from two trays per species (four trays in total). We sampled larvae every 48 h during stages L1–L4 to ensure distinct developmental stages. For adult females, we collected individuals 24 h after emergence, ensuring they had not taken a blood meal. We preserved each sampled individual in 100 µL of sterile phosphate-buffered saline (PBS) and stored them at −20°C.

### DNA extraction and quantitative real-time PCR

We extracted DNA from each individual using the Qiagen "DNeasy Blood and Tissue Kit" (Qiagen, Hilden, Germany) following the manufacturer's instructions and quantified it using the Qubit dsDNA HS Assay Kit (Thermo Fisher Scientific, Waltham, MA, USA, Invitrogen). We performed quantitative PCR (qPCR) on all individuals of both species (*Cx.*

quinquefasciatus and Cx. pipiens molestus). All qPCRs (45 cycles of 95°C for 10 s, 58°C for 20 s, and 65°C for 20 s) were performed on the Lightcycler LC480 real-time PCR instrument (Roche Diagnostics, Mannheim, Germany) and the SensiFAST SYBR No-ROX Kit (Bioline, Meridian Bioscience, London, UK), according to the manufacturer's instructions. We carried out each qPCR reaction in duplicate using a 6 µL mix (2.5 µL of SYBR No-ROX, 1 µL of DNA, each primer at 0.6 µM, adjusted to a total volume of 6 µL with nuclease-free water (Qiagen, Hilden, Germany).

We used specific primers (Table S1) to target the Wolbachia surface protein gene (wsp), plasmid pWCP gene (GP11), and the mosquito acetylcholinesterase gene (ace2), all present in a single copy and unique to each genomic entity. Serial dilutions of a synthetic plasmid (Eurofins Scientific SE, Luxembourg) including the genes GP11, wsp, and ace2 were placed in triplicate on each qPCR plate (four serial dilutions per plate) for each gene. These dilutions were performed in 10-fold increments, ensuring a reliable quantitative range for analysis. These served as internal controls to generate standard curves for each gene and each qPCR plate, mitigating any potential bias from variations between qPCR plates.

For each qPCR assay, we ensured that primer efficiency was close to 2 (100%) and that the standard curve slope remained near −3.33, in accordance with expected qPCR efficiency parameters. Additionally, we systematically checked the melting temperatures (Tm) of each amplicon to confirm the specificity of amplification across all samples. Fluorescence data were analyzed with the LightCycler480 software, which converted Cp values into concentrations (ng/µL) based on the standard curves. Samples with Cp values greater than 30 were excluded from the analysis.

We determined the relative proportions of each gene by normalizing the calculated concentrations of wsp and GP11 genes to gene ace2, allowing us to account for variations in DNA quantities between the samples.

## Statistical analyses

We performed all statistical analyses and data visualizations using R software (R Core Team, version 4.3.1, Vienna, Austria, 2023). We analyzed data from the two mosquito species independently, with each undergoing identical analyses. Linear models were used to investigate plasmid quantity (GP11/wsp ratio) and Wolbachia quantity (wsp/ace2 ratio) across developmental stages. Generalized linear mixed models (GLM, lme4 package) (47) were applied. Model assumptions (normality and homoscedasticity of residuals) were verified using the DHARMa (48) and Performance (49) packages. Parameters were analyzed under a normal distribution, and pairwise comparison tests were conducted to compare plasmid and Wolbachia quantities between stages.

We performed correlation analyses to examine the relationship between Wolbachia quantities and plasmid pWCP quantities at each developmental stage and for each individual. The choice between Pearson and Spearman correlation tests was determined by testing the raw data for normality within each developmental stage. We then applied either Student's t-tests (for normally distributed data) or Wilcoxon rank-sum tests (for non-normal data) to compare Wolbachia quantities between species at each developmental stage. Of note, correlation analyses are calculated on an individual basis while a comparison of dynamic analyses for Wolbachia and its plasmid pWCP focuses on the developmental stage level as a whole.

## RESULTS

### Wolbachia and pWCP plasmid abundance in Cx. pipiens molestus

We quantified the dynamics of Wolbachia and pWCP plasmid across the different developmental stages of Cx. pipiens molestus (Fig. 1A and B). For Wolbachia quantities (Fig. 1A), we observed an average bacterial copy number (wsp/ace2 ratio) ranging from 0.165 ± 0.213 (at the L3 stage) to 0.805 ± 0.474 (in adult females), with a significant increase at the pupal and adult female stages compared with other stages (P < 0.0001).

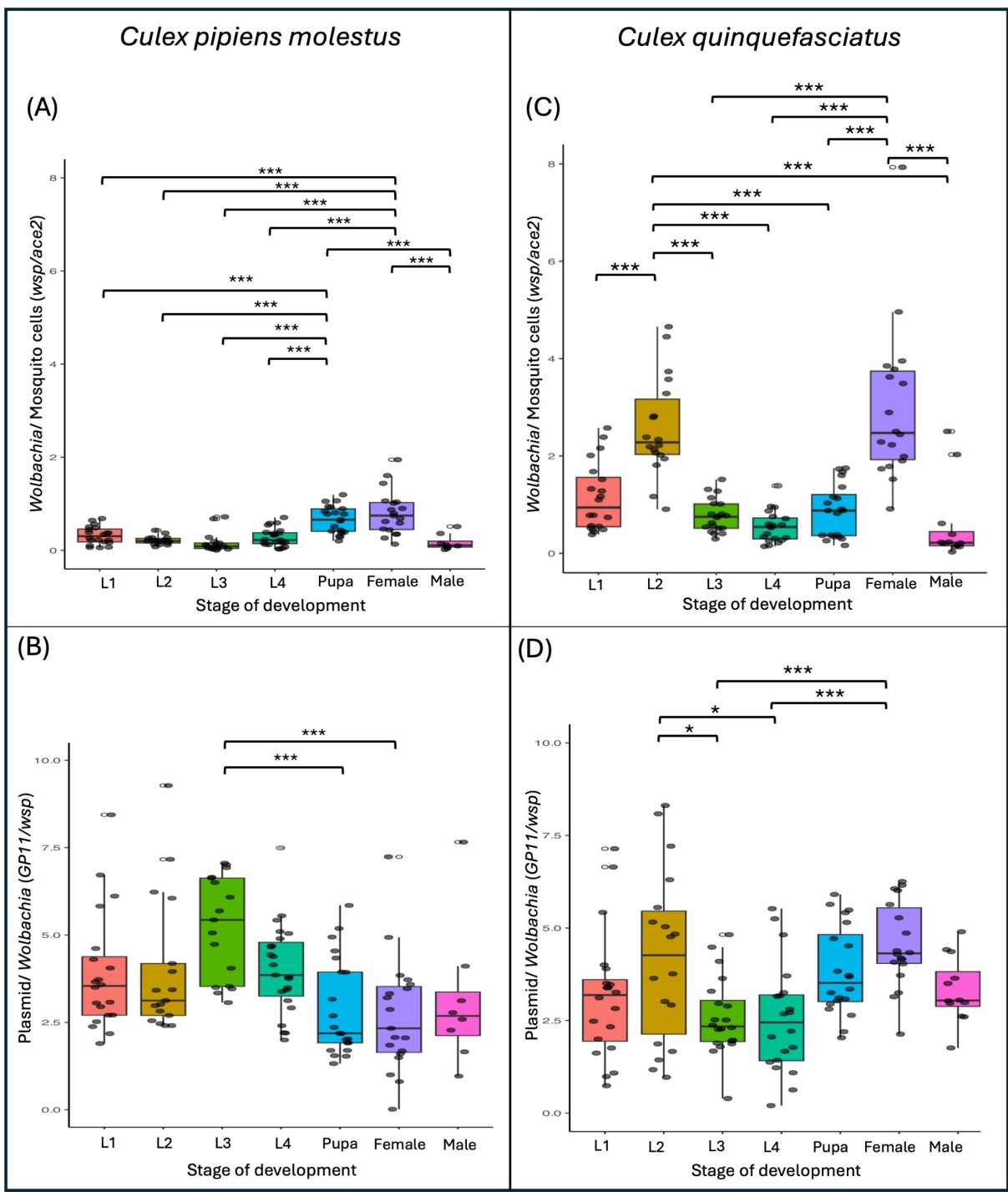

**FIG 1** Dynamics of plasmid and *Wolbachia* quantities during the life cycle of *Cx. pipiens molestus* and *Cx. quinquefasciatus* mosquitoes at the developmental stage level. The relative quantities of *Wolbachia* (*wsp* gene) are shown in (A) and (C) for *Cx. pipiens molestus* and *Cx. quinquefasciatus*, respectively, whereas the ones of the plasmid pWCP (*GP11*) are shown in (B) and (D) for each species. Boxplots indicate the median and interquartile range of the data. The gray dots represent individual values and the white circles next to it highlight the outliers identified in the data set. * indicates significance <0.05, ** indicates significance <0.01, and *** indicates significance <0.001.

For pWCP counts, the average ratio of the copy number of the plasmid by *Wolbachia* cell ranged from 2.69 ± 1.65 (in adult females) to 5.22 ± 1.48 (at the L3 larval stage), indicating that there are multiple copies of the plasmid in each cell. Pairwise comparisons adjusted using the Bonferroni method revealed a significant increase in plasmid

copy number by *Wolbachia* cell at the L3 stage, notably compared with adult females (SE = 0.539; *P* = 0.0002) and the pupal stage (SE = 0.527; *P* = 0.0005) (Fig. 1B).

Notably, we observed a significant positive correlation between the densities of pWCP and *Wolbachia* (*P* < 0.01) (Fig. 2A). A particularly strong slope and correlation were observed at the L3 larval stage (R = 0.995; *P* = 2.28E-16, Pearson correlation), indicating a higher number of plasmid copies per *Wolbachia* at this stage compared with the other developmental stages. Overall, slope values varied according to the developmental stage, indicating that plasmid copy numbers may change throughout development (Fig. 2A), but overall the plasmid co-replicates with the *Wolbachia* cells.

### *Wolbachia* and pWCP plasmid abundance in *Cx. quinquefasciatus*

In *Cx. quinquefasciatus* (Fig. 1C and D), *Wolbachia* copy number ranged from 0.545 ± 0.323 (at the L4 stage) to 2.99 ± 1.62 (in females), with a significant increase at the L2 and female stages compared with the other stages (*P* < 0.0001) (Fig. 1C). For the pWCP plasmid, copy numbers ranged from 2.54 ± 1.48 (at the L4 stage) to 4.55 ± 1.16 (in females) (Fig. 1D), which are similar to that of *Cx. pipiens molestus* and indicative of multiple plasmid copies per *Wolbachia* cell. Adjusted pairwise comparisons showed that plasmid quantities were higher in adult females and L2, significantly compared with the L3 (SE = 0.485; *P* = 0.0021; SE = 0.485; *P* = 0.0258) and L4 stages (SE = 0.485; *P* = 0.0013; SE = 0.485; *P* = 0.0170), respectively.

Correlation analysis between *Wolbachia* and plasmid densities revealed similarly positive correlations for all but the L2 larval development stage of *Cx. quinquefasciatus* (Fig. 2B) (R = −0.007; *P* = 0.97). At this particular stage, some individuals displayed a marked increase in *Wolbachia* without a concomitant increase in plasmid copies, unlike at other stages. Consequently, these results suggest that the plasmid co-replicates with the *Wolbachia* cells except during this larval stage (Fig. 2B).

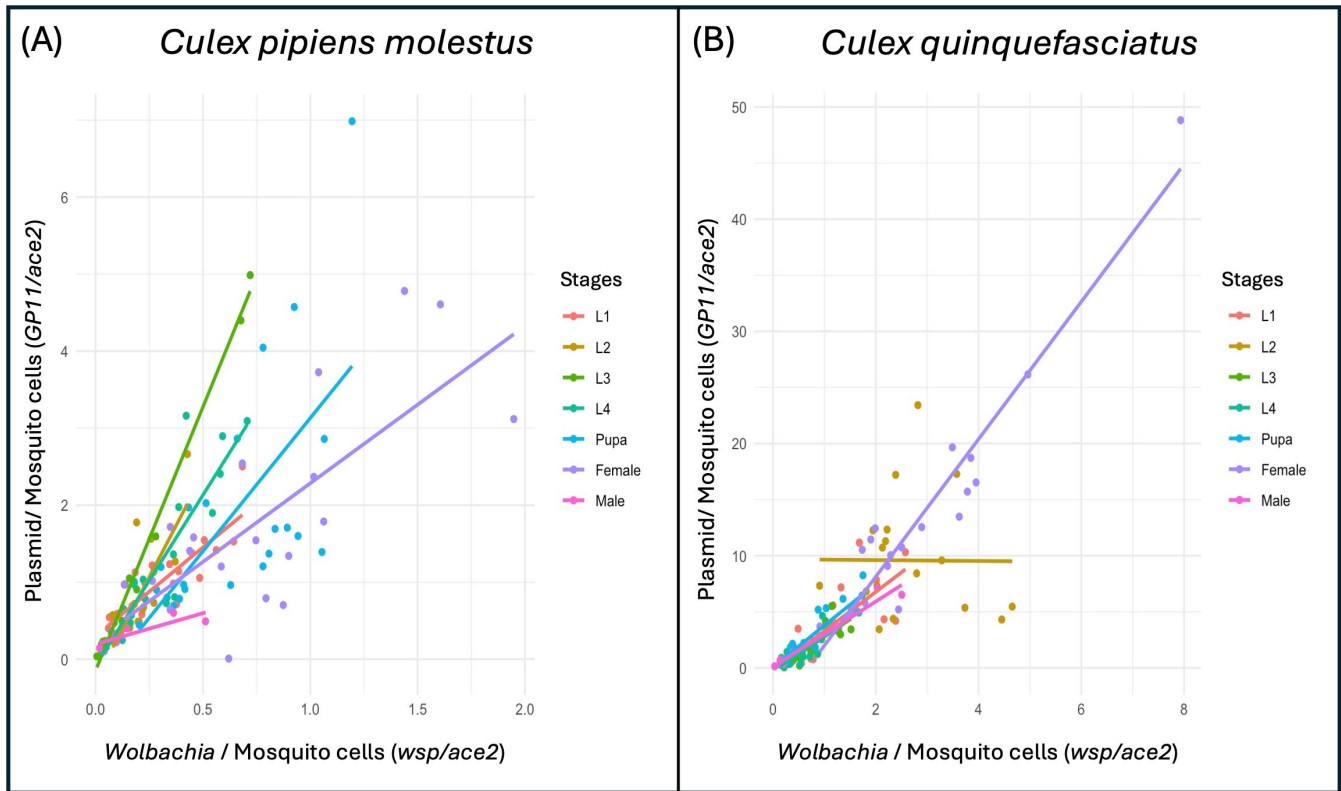

**FIG 2** Correlation analyses between *Wolbachia* and plasmid of *Cx. pipiens molestus* and *Cx. quinquefasciatus* mosquitoes at the individual level. The correlation analyses between *Wolbachia* and plasmid quantities are represented in (A) for *Cx. pipiens molestus* and in (B) for *Cx. quinquefasciatus*. In (A) and (B), all correlations are significant (<0.001), except for the L2 stage in (B).

## Comparison of *Wolbachia* quantities between *Cx. pipiens molestus* and *Cx. quinquefasciatus*

Next, we investigated the potential differences in *Wolbachia* quantities (*wsp/ace2* ratio) between *Cx. pipiens molestus* and *Cx. quinquefasciatus*. The results revealed significant differences between the two species at most stages. We observed a highly significant difference in *Wolbachia* density between species at the female ($W = 11$, $P = 2.21 \times 10^{-8}$), L1 ($W = 30$, $P = 4.07 \times 10^{-7}$), L2 ($t = -9.54$, $P = 5.19 \times 10^{-11}$), L3 ($W = 18$, $P = 2.01 \times 10^{-7}$), and L4 ($t = -3.35$, $P = 0.0017$) stages, with *Cx. quinquefasciatus* consistently showing higher densities than *Cx. pipiens molestus*. However, no significant differences in *Wolbachia* density were observed at the male ($P = 0.0691$) or pupal ($P = 0.3862$) stages.

## DISCUSSION

This study investigates the presence and dynamics of *Wolbachia* and pWCP plasmid across the developmental stages of *Cx. pipiens molestus* and *Cx. quinquefasciatus*. Our results show that both *Wolbachia* and its pWCP plasmid are present at all developmental stages, and for nearly all stages, the plasmid co-replicates with *Wolbachia* cells with roughly 2–5 copies per bacteria. Minor variations in counts or densities may reflect stage-specific as well as species-specific dynamics. Furthermore, although variations in the copy number of pWCP plasmid were mostly correlating with bacterial density, exceptions at the L2 larval stage of one species suggested a desynchronization of replication and complex interactions between *Wolbachia* and its associated mobile genetic element.

We observed differences in *Wolbachia* quantities between individuals at the same developmental stage within the same species. However, these variations were relatively minor, with *Wolbachia* levels varying up to 2-fold to 3-fold between individuals within the same stage. In contrast, variations observed in natural insect populations such as *Drosophila* can be much more substantial, reaching up to 20,000-fold differences between certain individuals (50). This low variability in our laboratory colonies suggests a relative stability of *Wolbachia* levels under controlled environmental conditions. Nonetheless, differences persist despite these stable conditions, reflecting the inherent variability of *Wolbachia* densities in infected hosts. Overall, intra-stage variations remained nevertheless lower than inter-stage variations.

The distinct *Wolbachia* dynamics observed between the two species—with increased bacterial abundance starting at the pupal and adult female stages in *Cx. pipiens molestus*, and at the adult female stage in *Cx. quinquefasciatus*—may be linked to differences in their reproductive biology. *Cx. pipiens molestus* is an autogenous species capable of reproducing without a prior blood meal, unlike *Cx. quinquefasciatus*, which is anautogenous and requires a blood meal to initiate ovarian development. In an autogenous species like *Cx. pipiens molestus*, the previtellogenic phase, which allows follicles to develop and become competent for fertilization, may begin earlier, potentially at the pupal stage (51). The release of juvenile hormone, necessary to progress beyond stage 1 of previtellogenic development and which induces an increase in bacterial abundance in the ovaries (28), may occur before emergence in autogenous species, unlike in anautogenous species, where it is released only after the first blood meal. Since *Wolbachia* is highly concentrated in the ovaries (52), the early development of ovaries in this species could explain the bacterium's increase observed at the pupal stage. A previous study reported an increase in *Wolbachia* levels at the adult female stage compared with the larval stages in the SLAB strain (*Cx. quinquefasciatus*), although only the L4 stage was sampled (53). A similar increase between the L4 and female stages was observed in the SLAB strain in this study, but an additional increase in *Wolbachia* was also detected at the L2 larval stage. These results suggest that either a stress-induced response or a physiological change occurring at this developmental stage would induce a sudden increase in the intracellular bacterium, although the exact causes and triggers remain to be determined. In both species, *Wolbachia* quantities were nevertheless higher in females than in males, a result consistent with previous observations in other species

such as *Drosophila simulans* (54, 55) and *Aedes albopictus* (56). Although *Wolbachia* is present in various tissues, it is primarily concentrated in the ovaries. The higher infection load in females can therefore be attributed to the significantly larger size of the ovaries compared with the testes.

When examining *Wolbachia* and plasmid quantities for each individual, we observed variations in pWCP plasmid copy number along the mosquito life cycles, as well as the differences between the two species. In *Cx. pipiens molestus*, variations in the pWCP plasmid were positively correlated with variations in *Wolbachia* levels at all developmental stages, indicating that when *Wolbachia* increase, the plasmid copy number also increases (Fig. 2A). This observed correlation differs from the previously reported inverse correlation between temperate phage WO and *Wolbachia* in *Nasonia* parasitoid wasps (57) whereby the phage can lyse *Wolbachia* cells during its replication stages to potentially invade new *Wolbachia* cells. A stronger correlation was observed at the L3 larval stage, suggesting a higher number of plasmid copies per *Wolbachia* at this specific stage. This increase in plasmid copy number may suggest a L3 stage-specific function of the plasmid in this species. Nevertheless, the plasmid overall co-replicates with the *Wolbachia* cells, suggesting a non-parasitic mobile element in this species.

In *Cx. quinquefasciatus*, a positive correlation between *Wolbachia* and pWCP copy numbers was observed across all developmental stages, except at the L2 stage, where a marked lack of correlation between *Wolbachia* and pWCP densities was detected (Fig. 2B). These results suggest a specific stress or physiological change at the L2 stage and that a notable increase in the intracellular bacterium load might cause at least a temporary desynchronization of plasmid replication. It would be pertinent to sample this L2 stage at different time points in *Cx. quinquefasciatus* with higher sample sizes to see if resynchronization of plasmid and *Wolbachia* replication occurs; this could better pinpoint the potential source of stress. It is possible that some *Wolbachia* cells may replicate too quickly for the plasmid to replicate and/or segregate properly, leading to the loss or to a lower number of the mobile genetic element in part of the *Wolbachia* population. However, the presence of two *RelBE* toxin–antitoxin (TA) systems in the pWCP plasmid may promote the stability of the mobile element by killing cells that lose the antitoxin component after segregation (58).

Generally, we observed that *Wolbachia* and pWCP plasmid quantities were consistently higher in *Cx. quinquefasciatus* than in the autogenous *Cx. pipiens molestus*. This observation aligns with previous studies that have shown interspecific differences between autogenous and anautogenous species in terms of *Wolbachia* densities, particularly in the testes, where strains such as Maclo (*Cx. quinquefasciatus*) exhibit higher *Wolbachia* densities than strains such as Tunis (*Cx. pipiens molestus*) (59). However, the density and variability differences observed between the two *Culex* species may result not only from their different life strategies (autogenous vs. anautogenous) but also from their distinct geographic origins (North America vs. France herein), as well as genetic differences between the *Wolbachia* groups in *Cx. pipiens* (*w*Pip) to which they belong. The SLAB strain (*w*Pip group III) (60) and the Celestine strain (*w*Pip group I or III) according to (60) show allelic differences in several genes (61), which could explain the variations in *Wolbachia* population dynamics and the abundance differences observed between the two species.

## Conclusion

The study of several mosquito species with distinct life cycles highlighted complex interactions and behaviors between pWCP and its hosts. Overall, variation of plasmid copy number per *Wolbachia* cell suggests largely stable multi-copy replication of the plasmid during mosquito development. The plasmid's copy number, stability, and transfer frequency could be influenced by the plasmid gene and regulatory elements, which appear to be highly conserved not only across species but also by host factors. The latter could affect the persistence and evolutionary dynamics of the plasmid within a population in different ways. Further studies to attempt localizing plasmid

accumulation in *Wolbachia* from different tissues (ovaries, midgut, etc.) using state-of-the-art microscopy tools, as well as expression studies of the plasmid's different genes, could help investigate the variations of the plasmid at a finer scale and better define its potential role in mosquito species.

## ACKNOWLEDGMENTS

We thank Angélique Porciani for helpful discussion on statistical analyses. This work was supported by the ERC RosaLind Starting Grant "948135" to JR and NIH R01 awards AI179743 and AI143725 to SRB. We thank the Vectopole platform (IRD, Montpellier, France) for providing technical support and for the rearing and maintenance of the mosquito populations. The Vectopole is a platform of the 'Vectopole Sud' Network and is part of the LabEx CeMEB (ANR-10-LABX-04–01).

A.B. designed and performed the experiments, analysed the data, prepared the figures, and wrote the manuscript. C.G. and J.T. contributed to the experiment design, coordinated laboratory experiments, and wrote the manuscript. S.B. and S.B. participated in the conception of this study and wrote the manuscript. B.T. coordinated data analysis and wrote the manuscript. J.R. conceived and coordinated this study, and wrote the manuscript.

## AUTHOR AFFILIATIONS

[1]Mivegec, Université de Montpellier, INRAE, CNRS, IRD, Montpellier, France
[2]Departments of Biology and Entomology, One Health Microbiome Center, Huck Institutes of the Life Sciences, Pennsylvania State University, University Park, Pennsylvania, USA

## AUTHOR ORCIDs

Seth R. Bordenstein  http://orcid.org/0000-0001-7346-0954
Sarah R. Bordenstein  http://orcid.org/0000-0001-6092-1950
Julie Reveillaud  http://orcid.org/0000-0002-2185-6583

## FUNDING

| Funder | Grant(s) | Author(s) |
|---|---|---|
| European Research Council | 948135 | Julie Reveillaud |

## AUTHOR CONTRIBUTIONS

Alice Brunner, Conceptualization, Formal analysis, Investigation, Methodology, Visualization, Writing – original draft | Camille Gauliard, Investigation, Methodology, Writing – review and editing | Jordan Tutagata, Investigation, Methodology, Writing – review and editing | Seth R. Bordenstein, Conceptualization, Investigation, Writing – review and editing | Sarah R. Bordenstein, Conceptualization, Investigation, Visualization, Writing – review and editing | Blandine Trouche, Conceptualization, Formal analysis, Investigation, Methodology, Supervision, Visualization, Writing – original draft | Julie Reveillaud, Conceptualization, Funding acquisition, Investigation, Methodology, Project administration, Supervision, Validation, Visualization, Writing – original draft

## DATA AVAILABILITY

Raw qPCR data for *Culex pipiens molestus* and *Culex quinquefasciatus* are available in Tables S2 and S3, respectively. Raw *Wolbachia* and pWCP quantities for species comparison are available in Table S4. A reproducible bioinformatics workflow (R code) used for this study is available in Supplementary Note 1.

## ADDITIONAL FILES

The following material is available online.

## Supplemental Material

**Data S1 (Spectrum00046-25-s0001.pdf).** R code.

**Supplemental material (Spectrum00046-25-s0002.docx).** Table S1; Supplemental table legends.

**Table S2 (Spectrum00046-25-s0003.xlsx).** Raw qPCR data for *Culex pipiens molestus*.

**Table S3 (Spectrum00046-25-s0004.xlsx).** Raw qPCR data for *Culex quinquefasciatus*.

**Table S4 (Spectrum00046-25-s0005.xlsx).** Raw *Wolbachia* and pWCP quantities for species comparison.

## Open Peer Review

**PEER REVIEW HISTORY (review-history.pdf).** An accounting of the reviewer comments and feedback.

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
