## [Reviewer comments · Microbiology Spectrum]

Microbiology Spectrum

Wolbachia and its pWCP plasmid show differential dynamics during the development of *Culex* mosquitoes

Alice Brunner, Camille Gaudiard, Jordan Tutagata, Seth Bordenstein, Sarah Bordenstein, Blandine Trouche, and Julie Reveillaud

Corresponding Author(s): Julie Reveillaud, Universite de Montpellier

Review Timeline:

Submission Date:	January 17, 2025
Editorial Decision:	February 11, 2025
Revision Received:	February 27, 2025
Accepted:	February 28, 2025

Editor: Jennifer Auchtung

Reviewer(s): Disclosure of reviewer identity is with reference to reviewer comments included in decision letter(s). The following individuals involved in review of your submission have agreed to reveal their identity: Marco Salgado (Reviewer #2)

Transaction Report:

DOI: <https://doi.org/10.1128/spectrum.00046-25>

Re: Spectrum00046-25 (Wolbachia and its pWCP plasmid show differential dynamics during the development of Culex mosquitoes)

Dear Dr. Julie Reveillaud:

Thank you for the privilege of reviewing your work. Below you will find my comments, instructions from the Spectrum editorial office, and the reviewer comments.

The reviewers have noted additional experiments and analyses that would be required to draw conclusions from this study. Please fully address these critiques in any future revision.

Revision Guidelines

Sincerely,
Jennifer Auchtung
Editor
Microbiology Spectrum

Reviewer #1 (Comments for the Author):

This study follows the ploidy of a plasmid, identified in some Wolbachia strains, throughout mosquito development. Although the concept is interesting, the data amount to simple qPCR experiments without multiple primers targeting the symbiont or the plasmid. This lack of robust rigor and the fact that the dataset is so small, decreased my enthusiasm for this work.

Major concerns:

The authors should not have used a single gene to identify copy numbers of these entities within the host. Additionally, the authors should most certainly have done a qPCR analysis using copy number, and not relative delta-delta Ct values as what they are trying to conclude is about copy numbers. What is the efficiency of each primer pair? This will make a big difference with regards to amplification.

It is unclear why the authors are normalizing to a host gene at all since each sample is a single individual and each individual DNA extraction is used in each qPCR with the same volume added to the reaction.

Reviewer #2 (Comments for the Author):

The manuscript of Brunner et al. reports the abundance of endosymbiotic *Wolbachia* as well as that of its plasmid pWCB across five developmental stages (L1-L4, and pupa) of two species of mosquitoes, *Culex pipiens molestus* and *Culex quinquefasciatus*, including analysis carried on both male and female adults. Abundance was estimated by quantitative PCR based on the ratios drawn from the quantity of selected markers genes: *Ace2* served as a marker for hosts (mosquitoes), *wsp* as a marker for *Wolbachia*, and *GP1* as a marker for the plasmid pWCB. The major conclusion from this study is that levels of infection by *Wolbachia* and the quantity of its plasmid pWCB vary not only between different developmental stages or between sexes, but also between species of mosquitoes. This is a simple study with a simple premise, but, as it stands, the experimental design does not provide enough grounds to support the conclusions and, despite its important message for ecologists, there are important lacunae in the manuscript that need to be addressed:

Major points:

- 1) Lack of proper description about the choice of marker genes. Why only one marker gene was chosen? For instance, to estimate the quantity of *Wolbachia*, why not include other marker genes such as those five genes used for Multi Locus Sequence Typing? Yet, concerning the choice of the marker genes *Ace2*, *wsp*, and *GP1*, are these genes stably expressed across all the conditions tested? This is extremely important for comparative purposes and must thus be included.
- 2) I have other concerns about the methodology, namely: i) How was the overall quality of the extracted RNA and which quantity of RNA was used to prepare cDNAs? ii) are we in conditions to rule out the possibility of primer dimer based on melting dissociation curves? iii) information about primer efficiency was not provided iv) in addition, I would encourage including information about the curves and the serial dilution used; v) why the cDNAs were not normalized across the preparations?
- 3) Line 283, this section of Results is not supported by any data either in the form of a Figure or Table.
- 4) Data availability. Gene sequences should be provided or, in case they are already available in public repositories, their accessions should be provided.

Minor points:

- 1) The Introduction needs a brief first paragraph about the importance of the study, if nothing else to include references to the statements mentioned in the "Importance" and "Abstract" sections.
- 2) Line 167: 10% sugar water? Please provide information about the reagents and concentrations used to prepare this solution.
- 3) Ensure that the general audience can easily read the information provided in the Supplementary Tables by translating it from French to English.

Point by Point response to Reviewers

Reviewer #1 (Comments for the Author):

This study follows the ploidy of a plasmid, identified in some *Wolbachia* strains, throughout mosquito development. Although the concept is interesting, the data amount to simple qPCR experiments without multiple primers targeting the symbiont or the plasmid. This lack of robust rigor and the fact that the dataset is so small, decreased my enthusiasm for this work.

Major concerns:

The authors should not have used a single gene to identify copy numbers of these entities within the host.

We sincerely appreciate the reviewer's comment and their concern regarding the use of a single gene to estimate copy numbers. In our study, we specifically targeted the plasmid pWCP and the *Wolbachia* genome using genes that are both unique to these entities and present in a single copy.

For the plasmid, we used the GP11 gene, which is unique to pWCP and exists in a single copy per plasmid unit. Since no other genes within pWCP would provide additional independent information regarding plasmid copy number, we believe targeting multiple genes would not enhance the robustness of our estimate. The detection of one copy of GP11 directly corresponds to the detection of one copy of pWCP, making it a reliable proxy for plasmid quantification. To further confirm this, we conducted qPCR analyses on three different plasmid genes—GP11, DnaB_2, and GP05 (some data are presented below)—and found that, for each sample, the Cp values and the associated concentrations were identical across all three targets. This result confirms that including multiple genes would yield redundant information and would not improve the accuracy of plasmid copy number estimation.

Similarly, for *Wolbachia*, we targeted the *wsp* gene, which is also unique to the bacterial genome and present in a single copy per bacterial cell. By normalizing its copy number to that of *ace-2*, a single-copy gene of the mosquito host, we obtain an accurate estimation of the *Wolbachia* to host ratio. Since additional *Wolbachia*-specific single-copy genes would provide redundant information, we believe that including multiple targets would not add significant value to the study.

We completed the following sentence to clarify this point: We used specific primers (**Supplementary Table 1**) to target the *Wolbachia* surface protein gene (*wsp*), plasmid pWCP gene (*GP11*), and the mosquito acetylcholinesterase gene (*ace2*), all present in single copy and unique to each genomic entity (New Line 203).

Raw data for qPCR analyses on three different plasmid genes:

Genes	Cp (triplicat)				Tm			
	Sample 1	Sample 2	Sample 3	Sample 4	Sample 1	Sample 2	Sample 3	Sample 4
GP11	22,88	16,94	12,45	13,82	83,44	83,58	83,53	82,93
	22,82	16,99	12,39	13,62	83,37	83,48	83,40	83,08
	22,75	17,29	12,40	13,72	83,30	83,47	83,51	82,82
RelB_2	22,80	17,39	12,66	13,96	81,61	82,06	82,04	81,45
	22,75	17,50	12,71	13,84	81,63	82,08	81,96	81,73
	22,82	17,45	12,46	13,90	81,63	82,03	82,09	81,48
GP04	22,93	17,34	12,76	13,88	80,83	80,98	80,96	80,64
	22,78	17,16	12,67	14,29	80,77	80,94	80,86	80,06
	22,76	17,53	12,84	13,97	80,77	80,87	80,93	80,06

Additionally, the authors should most certainly have done a qPCR analysis using copy number, and not relative delta-delta Ct values as what they are trying to conclude is about copy numbers.

We sincerely appreciate the reviewer's suggestion regarding qPCR analysis and the importance of using absolute copy numbers rather than relative $\Delta\Delta C_t$ values. However, we would like to clarify that we do not use the $\Delta\Delta C_t$ method in this study. Instead, we rely on standard curves to determine the absolute concentration of each target in our samples.

By using standard dilution series, we obtain the absolute copy number of each target gene in every sample. We then compute ratios to derive the plasmid copy number per *Wolbachia* cell and the *Wolbachia* copy number relative to mosquito cells, ensuring that our conclusions are indeed based on copy number quantification.

We hope this clarification addresses the reviewer's concern and appreciate their valuable feedback in refining our manuscript.

What is the efficiency of each primer pair? This will make a big difference with regards to amplification.

We appreciate the reviewer's concern regarding primer efficiency, as it is indeed a crucial factor in qPCR amplification. We would like to clarify that all our primers are highly efficient, and their specificity is systematically verified.

For each qPCR analysis, we ensure that the melting temperatures (Tm) of the amplicons match the expected values, confirming the specificity of each target gene. Additionally, the corresponding standard curves are rigorously assessed for each

gene. The slope and efficiency of these calibration curves consistently meet the optimal qPCR conditions, with slopes close to -3.33 and efficiencies close to 2 (i.e., 100% efficiency). The tenfold serial dilutions used in our standard curves show a consistent three-cycle difference between each dilution point, further confirming the reliability of our quantification.

We have now included this information in the manuscript to ensure clarity. “These dilutions were performed in tenfold increments, ensuring a robust dynamic range for quantification. These served as internal controls to generate standard curves for each gene and each qPCR plate, mitigating any potential bias from variations between qPCR plates. For each qPCR assay, we ensured that primer efficiency was close to 2 (100%) and that the standard curve slope remained near -3.33, in accordance with expected qPCR efficiency parameters. Additionally, we systematically checked the melting temperatures (T_m) of each amplicon to confirm the specificity of amplification across all samples” (New Line 206 to 213).

It is unclear why the authors are normalizing to a host gene at all since each sample is a single individual and each individual DNA extraction is used in each qPCR with the same volume added to the reaction.

In our study, each sample corresponds to a single individual, and our objective is to determine both the plasmid copy number per *Wolbachia* cell and the *Wolbachia* copy number per mosquito cell.

Although the same volume of extracted DNA (1 μ L) is added to each qPCR reaction, the total DNA concentration varies between individuals due to biological differences in *Wolbachia* density, plasmid content, and mosquito DNA yield. Direct comparisons of raw qPCR data without normalization would not account for these variations.

To ensure that our estimates are biologically meaningful, we calculate ratios rather than relying on absolute quantifications. By normalizing the number of copies of *wsp* (a single-copy gene in *Wolbachia*) to *ace-2* (a single-copy gene in the mosquito genome), we obtain an accurate estimate of the *Wolbachia*-to-host cell ratio. Similarly, by normalizing the copy number of GP11 (a single-copy gene in pWCP) to *wsp*, we determine the plasmid-to-*Wolbachia* ratio. These ratios allow us to compare plasmid and *Wolbachia* dynamics across individuals while accounting for variations in DNA input.

Reviewer #2 (Comments for the Author):

The manuscript of Brunner et al. reports the abundance of endosymbiotic *Wolbachia* as well as that of its plasmid pWCB across five developmental stages (L1-L4, and pupa) of two species of mosquitoes, *Culex pipiens molestus* and *Culex quinquefasciatus*, including analysis carried on both male and female adults. Abundance was estimated by quantitative PCR based on the ratios drawn from the quantity of selected markers genes: *Ace2* served as a marker for hosts (mosquitoes), *wsp* as a marker for *Wolbachia*, and GP1 as a marker for the plasmid pWCB. The major conclusion from this study is that levels of infection by *Wolbachia* and the quantity of its plasmid pWCB vary not only between different developmental stages

or between sexes, but also between species of mosquitoes. This is a simple study with a simple premise, but, as it stands, the experimental design does not provide enough grounds to support the conclusions and, despite its important message for ecologists, there are important lacunae in the manuscript that need to be addressed:

Major points:

1) Lack of proper description about the choice of marker genes. Why only one marker gene was chosen? For instance, to estimate the quantity of *Wolbachia*, why not include other marker genes such as those five genes used for Multi Locus Sequence Typing?

We thank the reviewer for his comment. As addressed in our response to reviewer #1, our rationale for selecting a single marker gene for both *Wolbachia* and the plasmid was the following: we specifically targeted genes that are both unique to their respective entities and present in a single copy, ensuring a direct estimation of copy numbers. Additionally, we conducted qPCR analyses on three different plasmid genes (GP11, DnaB_2, and GP05) and found identical Cp values and associated concentrations across all three, demonstrating that using multiple targets would be redundant. For *Wolbachia*, the *wsp* gene was chosen because it is single-copy and species-specific, making it a reliable proxy for bacterial quantification.

We refer the reviewer to our detailed response above for further clarification and sincerely appreciate their feedback.

Yet, concerning the choice of the marker genes *Ace2*, *wsp*, and GP1, are these genes stably expressed across all the conditions tested? This is extremely important for comparative purposes and must thus be included.

We would like to clarify that this study focuses on DNA quantification (copy number estimation), not gene expression. Since we are analyzing genomic DNA (gDNA) rather than RNA, there is no variability in gene expression to consider. The genes *ace-2*, *wsp*, and GP11 are all present as single-copy genes in the mosquito genome, *Wolbachia*, and plasmid pWCP, respectively, and their stability is therefore intrinsic, as they are genetically encoded and not subject to transcriptional regulation.

2) I have other concerns about the methodology, namely: i) How was the overall quality of the extracted RNA and which quantity of RNA was used to prepare cDNAs?

We appreciate the reviewer's comment; however, we would like to clarify that this study does not involve RNA analysis at any point. We exclusively focused on DNA quantification to follow pWCP copy number, and no RNA extraction, cDNA synthesis, nor gene expression analysis was performed. RNA is not mentioned in the current manuscript and is beyond the scope of this study although pWCP expression analyses represent really interesting perspectives.

ii) are we in conditions to rule out the possibility of primer dimer based on melting dissociation curves?

The specificity of each primer pair was thoroughly validated, and melting dissociation curves were systematically analyzed for all qPCR reactions. No primer dimers were detected in any sample, as all dissociation curves showed a single, well-defined peak corresponding to the expected amplicon. We specified in the manuscript “Additionally, we systematically checked the melting temperatures (T_m) of each amplicon to confirm the specificity of amplification across all samples.” (New Line 213)

iii) information about primer efficiency was not provided

As addressed in our response to reviewer #1, we provided details on primer efficiency, including validation through standard curves, which consistently showed slopes close to -3.33 and efficiencies near 100%. These results confirm that all primers used in this study meet the expected qPCR efficiency criteria.

We have now included this information in the manuscript to ensure clarity. “For each qPCR assay, we ensured that primer efficiency was close to 2 (100%) and that the standard curve slope remained near -3.33, in accordance with expected qPCR efficiency parameters.” (New Line 210 to 212).

iv) in addition, I would encourage including information about the curves and the serial dilution used;

Thank you for your suggestion. We have added the information about the serial dilution in the manuscript. “These dilutions were performed in tenfold increments, ensuring a robust dynamic range for quantification.” (New Line 206-207).

v) why the cDNAs were not normalized across the preparations?

This study does not involve cDNA analysis, as we focus exclusively on DNA quantification rather than RNA expression. The normalization of DNA was performed after qPCR analysis using ratio calculations, ensuring reliable comparisons across samples.

Pre-normalizing DNA concentrations before qPCR was not implemented due to the large variability in DNA yield across developmental stages (e.g., between a first-instar larva and an adult female). Standardizing DNA inputs across all individuals would have required setting the whole dataset to the minimum concentration and therefore excessive dilution of samples with naturally high DNA concentrations, potentially leading to a loss of detection sensitivity for most samples. Instead, post-quantification normalization using calculated ratios was considered a robust and more appropriate approach for this study.

3) Line 283, this section of Results is not supported by any data either in the form of a Figure or Table.

We thank the reviewer for his note. These results are supported by Figure 2B (please see L2, brown color). We added this information in the manuscript.

4) Data availability. Gene sequences should be provided or, in case they are already

available in public repositories, their accessions should be provided.

Thank you for your comment. We added in supplementary Table 1 pWCP sequence for the custom-designed primers GP11F and GP11R. The remaining primers used in our study were selected based on alignment with reference sequences available in public databases, but they were not directly designed from these references and we could therefore not provide unique Accession numbers. Gene sequence for these primers are, to the best of our knowledge, not provided in the original articles.

Minor points:

1) The Introduction needs a brief first paragraph about the importance of the study, if nothing else to include references to the statements mentioned in the "Importance" and "Abstract" sections.

Thank you for your suggestion. We added a brief introductory paragraph highlighting the importance of the study and incorporating relevant references, as suggested: "However, despite the importance of pWCP for *Wolbachia* in *Culex*, much remains unknown about its replication mode, behavior and interaction with its host (New line 153 to 155).

2) Line 167: 10% sugar water? Please provide information about the reagents and concentrations used to prepare this solution.

The 10% sugar water solution refers to a mixture containing 10 g of white sugar dissolved in 100 mL of filtered water, meaning that the final concentration is 10% (w/v). This was clarified in the manuscript (New line 173).

3) Ensure that the general audience can easily read the information provided in the Supplementary Tables by translating it from French to English.

Thank you for your remark. We translated the Supplementary Tables into English to ensure clarity for general audience.

Re: Spectrum00046-25R1 (Wolbachia and its pWCP plasmid show differential dynamics during the development of Culex mosquitoes)

Dear Dr. Julie Reveillaud:

Your manuscript has been accepted, and I am forwarding it to the ASM production staff for publication. Your paper will first be checked to make sure all elements meet the technical requirements. ASM staff will contact you if anything needs to be revised before copyediting and production can begin. Otherwise, you will be notified when your proofs are ready to be viewed.

Sincerely,
Jennifer Auchtung
Editor
Microbiology Spectrum